# Comprehensive Dental Treatment under General Anesthesia Improves Mastication Capability in Children with Early Childhood Caries—A One-Year Follow-Up Study

**DOI:** 10.3390/ijerph20010677

**Published:** 2022-12-30

**Authors:** Natacha Linas, Marie-Agnès Peyron, Pierre-Yves Cousson, Nicolas Decerle, Martine Hennequin, Caroline Eschevins, Emmanuel Nicolas, Valérie Collado

**Affiliations:** 1Centre de Recherche en Odontologie Clinique (CROC), Université Clermont Auvergne, F-63000 Clermont-Ferrand, France; 2Service d’Odontologie, CHU Clermont-Ferrand, F-63003 Clermont-Ferrand, France; 3CRNH Auvergne, Human Nutrition Unit (UNH), Institut National de Recherche pour l’Agriculture, l’Alimentation et l’Environnement (INRAE), Université Clermont Auvergne, F-63000 Clermont-Ferrand, France

**Keywords:** children, dental caries, dental treatment, mastication, particle size, general anesthesia

## Abstract

Background: Using the granulometry of ready-to-swallow food boluses, this study investigated the evolution of masticatory capability of children with Early Childhood Caries (ECC) after comprehensive dental treatment under general anesthesia (GA). Methods: Sixteen children with ECC were assessed before and over one year after dental treatment under GA, in comparison with 12 children with a Healthy Oral State (HOS). Oral health criteria, quality of life, body mass index, and frequency of orofacial dysfunctions were recorded. Masticatory kinematic parameters and median food bolus particle size (D50) at swallowing were assessed while masticating raw carrot (CAR), cheese (CHS), and breakfast cereals (CER). The impact of posterior teeth extractions was analyzed. Results: Quality of life and orofacial functions improved after dental treatment. Chewing frequency for all three foods increased without reaching the values of children with HOS, while D50 values for CAR and CHS decreased. After one year, children with posterior teeth extractions exhibited higher D50 values for CAR and CHS than children with only conservative treatment. One third of children with ECC were overweight or obese. Conclusions: Comprehensive dental treatment improved children’s mastication, and their BMI subsequently increased. Links between mastication and nutrition should be investigated further in children.

## 1. Introduction

Early Childhood Caries (ECC) is widely recognized as a significant public health issue that especially affects children from socioeconomically disadvantaged families, with early exposure to high sugar intake through specific dietary patterns and drinking habits [1]. This disease is characterized by the presence of “one or more decayed, missing or filled tooth surfaces (due to decay) in any primary teeth in a child of 71 months of age or younger” [2]. In 2021, a survey using the WHO criteria estimated that this disease affected approximately one in two children worldwide, but with a very heterogeneous distribution [3]. In 2015, an estimated 573 million children had untreated dental caries in primary teeth and this prevalence has remained relatively unchanged for 30 years [3,4]. It is well known that this disease negatively affects children and families’ quality of life [5,6]. Furthermore, it has negative nutritional consequences. The risk of iron deficiency anemia and the risk of ferritin, vitamin (especially vitamin D), and calcium deficiencies is elevated [7,8,9,10,11], and the height–weight status of affected children is also altered [12,13,14]. Feeding difficulties are often cited in quality of life studies conducted in this population [5,6,15]. More recently, some studies have shown that, in spite of kinematic behavioral adaptations to their oral health alteration (use of infantile praxes, reduction of masticatory frequency) [16,17], children with ECC ended up with altered masticatory capability, meaning they swallowed insufficiently comminuted food boluses [18]. These difficulties may have consequences on subsequent nutrient absorption.

Treatment under general anesthesia (GA) is often required in children with ECC due to the complexity of the multiple dental procedures needed and their young age [19,20]. Children and families’ quality of life improve significantly after treatment under GA, as many studies have shown [15,21,22,23,24]. However, the impact of such one-time treatment on children’s mastication and more globally on their nutrition has not been assessed. It was also shown that ferritin, vitamin D, C-reactive protein (CRP), and insulin-like growth factor-1 (IGF-1) levels improved after dental treatment [24]. Some studies conducted in groups of children with ECC suggested that mean growth parameter values tended to return to normal after dental treatment [24,25,26]. However, the distribution of individual height-and-weight status according to standard childhood growth charts (i.e., whether children BMI was within, below, or above the normal range) was not described and the impact of potential persistent masticatory deficiencies after oral rehabilitation on the evolution of children’s height-and-weight status is still unknown.

In studies performing a subjective assessment of oral health-related quality of life, parents and children self-reported an improvement of mastication after dental treatment under GA [15,23,24,26]. However, it has been suggested, using the Nordic Orofacial Test Screening (NOT-S), that mastication dysfunction could persist to some extent three months after dental rehabilitation [21]. Recently, one study assessed the evolution of several masticatory parameters after treatment of ECC [27]. Children’s kinematic chewing criteria (chews/gram and chewing time/gram) were analyzed during the mastication of six real foods of different textures, and their mixing ability was assessed using dual-colored chewing gums. The authors concluded that children completed more chews to eat hard foods after dental treatment (Froot Loop cereals and peanuts). In contrast, the mixing ability assessed after a given number of chews on a bi-colored chewing gum was not globally modified three months after treatment. However, these data cannot reflect children’s mastication capability while consuming foods in a nutritive way [18]. Mastication capability depends on dental state, biting force, salivary flow, muscular activity, and sensorimotor control [28] and its development may be influenced by any condition modifying proprioceptive information from teeth during early childhood. Indeed, it is still unknown if the mastication capability of children with ECC (or any other oral health alteration) improves after dental treatment, as no follow-up granulometry study has been conducted in this population. Professionals may wonder if the complete normalization of the masticatory function can be expected after treatment under GA, or if an additional intervention should be implemented.

This study is the first to assess, over a one-year period, the evolution of the masticatory capability of children with ECC after comprehensive dental treatment in comparison with that of children from the same age range with a healthy oral state. The impact of such treatment on masticatory behavior, child BMI, oral health related quality of life, orofacial dysfunction frequency, and feeding habits at home was also assessed.

## 2. Material and Methods

### 2.1. Study Design

This prospective observational study was conducted in the special dental care unit of the University Hospital of Clermont-Ferrand (FRANCE) over a 4-year period (2016–2020) at the Center for Clinical Research in Odontology (CROC-UR4847). It was designed in accordance with Good Clinical Practice and with ethical standards (ref ICH expert working group 1996). It was approved by the local Ethical Committee (CECIC, 2010/06; IRB Number 5044). Children and parents, or legal representatives, were informed (oral and written information) about data confidentiality, the design and goals of the study, and the potential benefits and constraints related to their participation. A consent form was signed by the participants and their legal representatives.

### 2.2. Participants

Children from 36 to 71 months of age, without developmental disorders, with strict deciduous dentition at inclusion time, with Early Childhood Caries (ECC), and for whom the treatment was planned under general anesthesia were included in the study and formed the “Group with ECC”. The treatment plan under GA was as conservative as possible. Extractions were performed only if teeth exhibited destruction of the pulp chamber floor, a history of cellulitis, or in the case of extensive root or alveolar bone resorption. Three follow-up appointments were planned after dental treatment under GA. The analysis was conducted per protocol, thus inclusion in the Group with ECC implied that children had to attend all their appointments during the follow-up period.

The “Healthy Oral State group” (Group with HOS) was composed of children from the same age range, without any oral problems, and visiting the dental unit for their routine dental examination during the study period. Inclusion in this “Group with HOS” implied that children had to attend a second examination at least 6 months after the first one.

The number of subjects needed was estimated from a preliminary pilot study analyzing masticatory capability (via the mean D50 value for raw carrot (D50 CAR) before and 12 months after dental treatment under GA in eight children with ECC. The mean D50 CAR value was higher before treatment than 12 months after treatment (4.75 mm versus 3.43 mm, respectively) (SD = 1.04). Calculations were based on this difference for a continuous criterion with independent values and indicated the need for at least 11 subjects for each group (α = 5%, β = 10%, epiR package 0.9–30).

As children could refuse certain food samples, leading to missing data, data analyses were performed on 16 children in the Group with ECC (nine girls, seven boys) and 12 children in the Group with HOS (three girls, nine boys). In the Group with ECC, 38 children were included before treatment, but only 20 came to the 1–3 month follow-up appointment, 17 to the 6-month follow-up, and 16 to the 12-month follow-up. In the Group with HOS, 22 children were included in the first examination but only 12 came back for the second examination. Both groups were similar in terms of gender (Fisher’s exact test, NS).

### 2.3. Experimental Design and Data Collection

The participants in the Group with ECC were assessed before treatment (T0), and after treatment, at the 1–3 month follow-up (T1), at the 6-month follow-up (T2), and at the 12-month follow-up (T3). The participants in the Group with HOS were assessed on their first (T0) and second (T3) routine examination, which were at least 6 months apart.

Two investigators (dental practitioners) collected the descriptive variables and study criteria.

Sessions were planned between meals, in the mid-morning or mid-afternoon. Data relating to general health [age, height, weight, and body mass index (BMI)], oral health-related quality of life, orofacial dysfunctions, and feeding habits at home were collected in an interview with both the parents/caregivers and the child. Oral health criteria (oral state and occlusal parameters) were then gathered during the children’s clinical examinations. All the study criteria will be detailed in Section 2.4.

Finally, mastication tests aimed at gathering masticatory behavior data (food refusals, kinematic parameters, and impairment of muscular function) and mastication capability data (bolus granulometry at the time of swallowing determined after sieving) were performed. They were organized and performed as described in Linas et al. (2020) [16]. For each child, 12 calibrated food samples were prepared: 4 samples of raw carrot (CAR) and cheese (CHS) (cylindrical sample diameter 2 cm/thickness 1 cm, 3 g ± 10% respectively of raw carrot and Emmental Coeur de Meule cheese (Président^®^) and 4 samples of cereals (CER) (1 g ± 10% Fitness^®^ original breakfast cereal (Nestlé^®^). The children were encouraged to express any difficulty upon refusal or uneasiness with the food samples, and the tests were stopped at any time. Each subject was invited to chew and swallow two samples of each tested food. These two first masticatory sequences allowed live recording of individual swallowing thresholds (number of cycles needed before swallowing: Nc). For the next two samples of each food, subjects were stopped at the predetermined individual swallowing threshold (Nc) and the food bolus was collected and frozen (for at least 24 h at −18 °C) for further granulometric analysis by manual sieving as described in Linas et al. (2020) [16]. The weight of the particles retained by each of the eight sieves was recorded in order to draw a cumulative curve of particle mass falling through each sieve.

For the Group with ECC, GA took place on average 8.6 ± 5.6 months after the first assessment and on average 22.4 ± 5.7 months passed between the first and the last assessments. In the Group with HOS, on average 23.5 ± 14.1 months passed between the first and the last assessment. Figure 1 presents the global experimental procedure and the different period times between assessments.

### 2.4. Study Evaluation Criteria

#### 2.4.1. General Health-Related Criteria

Age, height, and weight were recorded for all the children, and the individual BMI was calculated using the following formula: BMI= $\frac{\mathrm{weight}\;\left(\mathrm{kg}\right)}{{\left(\mathrm{height}\;\left(m\right)\right)}^{2}}$.

The resulting BMI value obtained for each child was then positioned on the corresponding growth chart according to age and gender. The children were then included in one of the following categories described by the International Obesity Task Force (IOTF) [29]: underweight (BMI < IOTF 17), healthy weight (IOTF 17 ≥ BMI ≤ IOTF 25), overweight (IOTF 25 > BMI ≤ IOTF 30), and obese (IOTF 30 < BMI).

#### 2.4.2. Oral Health Criteria

Children’s carious status was assessed using the dmft/DMFT index (decayed, missing, filled teeth index in deciduous/PERMANENT dentition) [30]. The clinical consequences of caries were assessed using the PUFA index (visible pulp, ulceration of the oral mucosa due to root fragments, fistula, and abscesses) [31,32].

The number and type of dental treatments performed under GA were noted for children with ECC. For further analysis, children with ECC were separated in two groups, whether they had at least one posterior tooth extracted or none.

In addition, the number of permanent teeth present on the dental arches was noted.

The presence of orofacial dysmorphology was controlled in both groups at the different evaluation times, as it could influence mastication [33]. It was assessed using different occlusal deviations of the “dental health” component of the Index of Orthodontic Treatment Need (IOTN), categorized into: dento-maxillary disharmony, eruption or number anomalies, and sagittal, vertical, and transverse dimension abnormalities [34].

#### 2.4.3. Oral Health-Related Quality of Life

The impact of oral health on children’s quality of life was investigated using a validated French version of the early childhood oral health impact scale (ECOHIS) based on the parental rating of 13 items [35]. The 13 items were gathered in 6 domains also classified in 2 main parts: the child impact section (child symptoms, child functions, child psychology, and child self-image and social interactions) and the family impact section (parental distress and family function). For each item, a simple Likert-type scale ranging from “Never” (=0) to “Very often” (=4) was applied. An additional answer, “Do not know” is also available and scored as zero. The total score ranged from 0 to 52, with a higher score meaning a higher negative impact of oral health on quality of life.

#### 2.4.4. Orofacial Dysfunction Frequency

The frequency of orofacial dysfunction was assessed using the French version of the Nordic Orofacial Test-Screening (NOT-S) [36]. This test is divided in a structured interview and a clinical examination each exploring six functional domains (I to VI: “sensory function”, “breathing”, “habits”, “chewing and swallowing”, “drooling”, and “dryness of the mouth” for the structured interview and 1 to 6: “face at rest”, “nose breathing”, “facial expression”, “masticatory muscle and jaw function”, “oral motor function”, and “speech” for the clinical exam). Each domain was scored one point, even if several items within the domain are noted positively. The total score ranged from 0 to 12, with a higher score meaning a higher frequency of orofacial dysfunction.

#### 2.4.5. Feeding Habits at Home

The number of children exhibiting food refusals and/or texture adaptations, defined as adaptations of food texture (cooking, cutting into small pieces, mincing, etc.), at home were recorded for each of the seven food categories described in the National Program on Nutrition and Health food guide [37]. Refusals were recorded when the child refused at least one food within the food category.

#### 2.4.6. Masticatory Behavior

##### Food Refusals during Mastication Tests

For each tested food, the number of children displaying food sample refusals were recorded and the reasons for refusal were sought and classified into “Don’t like”, “Don’t know-Never tried”, “Painful”, or “Too difficult to eat”.

##### Kinematic Parameters

For each sequence of mastication, the following kinematic parameters were recorded:-Chewing time (Ti, seconds): time between the moment the food was placed into the mouth until the last food bolus manipulation, just before complete swallowing;-Number of masticatory cycles (Nc): number of chewing strokes during the chewing time period, with or without lip closure, corresponding to biting movements (tongue and perioral muscle manipulation movements were not counted);-Chewing frequency (Fq = Nc/Ti): calculated ratio between the number of masticatory cycles and chewing time.

##### Impairment of Muscular Function

Two trained investigators assessed the impairment of muscular function on mastication sequences recorded by video using a clinical tool developed by speech therapists [38]. As described in a previous study, the muscular function “predominant use of tongue to manage food” was added to the five items of the scale (including impaired incision, lip incompetence, unilateral masticatory pattern, impaired masticatory movements, and predominant use of perioral muscles) [16]. For each item, it was noted if the impairment was present or not.

#### 2.4.7. Mastication Capability

The mastication capability was assessed by food bolus granulometry analysis. Food bolus granulometry was expressed as the median particle size value (D50 value) of each food bolus collected at the moment of swallowing (Nc). For each sample, a cumulative curve was drawn from the particle mass passing through each sieve. The D50 value corresponds to the theoretical sieve size allowing 50% of the particle mass to pass through. A higher D50 shows a greater proportion of large particles in the food bolus.

### 2.5. Statistical Analysis

SPSS^®^ (IBM, v25) software was used to conduct the statistical analysis. The significance threshold value was set at *p* ≤ 0.05. The analysis strategy was first aimed at assessing the evolution of the parameters studied within the same group over time (comparing T0 versus T1, T0 versus T2 and T0 versus T3 for the Group with ECC and T0 versus T3 for the Group with HOS). Second, the analysis compared data between both groups of children at T0 and T3. Finally, the impact of posterior teeth extractions on chewing frequency and the D50 values was assessed for the Group with ECC.

#### 2.5.1. Evolution between Evaluation Times for Each Group of Children

Within the Group with ECC, the distribution of the BMI categories was compared between T0 and T3 using Wilcoxon tests. The ECOHIS and NOT-S scores, masticatory kinematic parameters, and D50 values were compared between T0 and the other evaluation times (T0 versus T1, T0 versus T2, and T0 versus T3) using Dunnett *t*-tests. The number of children reporting food refusals and texture adaptations at home, and refusing food samples during the mastication tests were compared between T0 and T3 using McNemar tests.

Within the Group with HOS, ECOHIS and NOT-S scores, masticatory kinematic parameters and D50 values were compared between T0 and T3 using Student’s *t*-tests.

#### 2.5.2. Comparison between Both Groups of Children at the Different Evaluation Times

Student’s *t*-tests were used to compare general and oral health data between the groups at the two evaluation times, except for gender and the presence of oral dysmorphologies for which Fisher exact tests were used. The comparison of ECOHIS and NOT-S scores, masticatory kinematic parameters and D50 values was performed using Student’s *t*-tests. Finally, Fisher’s exact tests were used to assess the comparison of the number of children exhibiting food refusals and texture adaptations at home, and exhibiting food refusals during the mastication tests in both groups of children.

#### 2.5.3. Impact of Posterior Teeth Extractions in Children with ECC

Chewing frequency and D50 values were compared at T0 and T3 in children with and without extractions of posterior teeth performed under GA, using Student’s *t*-tests.

## 3. Results

### 3.1. General Health-Related Characteristics

#### 3.1.1. Age, Weight and Height

Both groups were similar in terms of age, weight, and height at the time of inclusion (T0) and at the last evaluation time (T3). Table 1 presents the general health data recorded throughout the study in both groups of children.

#### 3.1.2. Body Mass Index

At T0, one child with ECC was underweight and one was overweight; the 14 others were in the healthy BMI range. Twelve months after GA, two children became overweight and three children reached the threshold for obesity. Between T0 and T3, two children underwent a normalization of their BMI (one from the underweight to the healthy category and one from the overweight to the healthy category), five children entered a pathological BMI category (two from the healthy range to overweight and three from the healthy range to obese) and nine children stayed in the healthy range. The BMI of children with Healthy Oral State remained in the healthy range. The difference of distribution within the BMI categories between T0 and T3 was significant (Wilcoxon test, *p* ≤ 0.05).

### 3.2. Oral Health Criteria

For children with ECC, the mean dmft score at T0 was 12.3 ± 4.6. The number of permanent first molars increased throughout the study. During GA, on average 3.5 ± 2.71 extractions and 8.19 ± 2.66 conservative treatments were performed. All data are presented in Table 2. The frequency of orofacial dysmorphologies was not significantly different between groups of children.

### 3.3. Oral Health-Related Quality of Life (ECOHIS)

The global mean ECOHIS score decreased significantly in children with ECC, after GA at T1 (Dunnett *t*-test, *p* ≤ 0.001), and in children with HOS between T0 and T3 (Student’s *t*-test, *p* ≤ 0.01). However, it remained significantly higher in children with ECC at both times (13.88 ± 5.1 vs. 0.92 ± 1.24 at T0 and 2.69 ± 2.18 vs. 0 at T3, respectively) (Student’s *t*-test, *p* ≤ 0.001). In particular, the “Difficulty eating” item score remained higher in children with ECC compared to children with HOS at T0 (1.56 ± 1.55 (0–4) vs. 0; *p ≤* 0.001) and T3 (0.56 ± 1.03 (0–3) vs. 0; *p ≤* 0.05).

### 3.4. Orofacial Dysfunction (NOT-S)

The NOT-S mean global score decreased immediately after treatment (T1) in children with ECC. The difference was significant between T0 (2.63 ± 1.45; min–max: 0–5) and, T1 (1.75 ± 1.61; min–max: 0–5), T2 (1.75 ± 1.13; min–max: 0–4) (Dunnett *t*-tests; *p* ≤ 0.05), and T3 (1.63 ± 1.26; min–max: 0–3) (*p* ≤ 0.01), respectively, unlike in children with HOS over the course of the study (at T0: 1.17 ± 1.11; min–max: 0–3; at T3: 0.83 ± 0.83; min–max: 0–2) (Student’s *t*-test, NS). The intergroup comparison showed higher NOT-S mean global scores for children with ECC than for children with HOS at T0 and at T3 (Student’s *t*-test; *p* ≤ 0.001 and *p* ≤ 0.01, respectively).

Domain-by-domain dysfunction frequencies are shown in Figure 2.

### 3.5. Feeding Habits at Home

From T0 to T3, fewer children with ECC reported food refusals and food texture adaptations at home. This decrease was not significant except for texture adaptation for meat (9/16 at T0 versus 2/16 at T3) (*p* ≤ 0.05; McNemar test). The higher frequencies of food refusals and texture adaptations were observed for meat. No child with HOS reported needing texture adaptations. Food refusals at home concerned fruits, vegetables, and dairy products.

The frequency of food refusals and texture adaptations in children with ECC tended to be higher than in children with HOS. The difference between groups at T0 was significant for meat refusals (6 children with ECC vs. 0 children with HOS) (*p* ≤ 0.05; Fisher’s exact test), for texture adaptations for meat (9 vs. 0) (*p* ≤ 0.01; Fisher’s test), and for fruits and vegetables (5 vs. 0) (*p* ≤ 0.05; Fisher’s exact test). At T3, there was no statistical difference between the groups.

### 3.6. Masticatory Behavior

#### 3.6.1. Food Refusals during the Mastication Tests

Refusals were analyzed by observing the number of children refusing each tested food. In this study, when the children refused a food, they systematically rejected all the samples of this given food. In children with ECC, the number of food refusals for each independently tested food (CAR: carrot; CHS: cheese; and CER: cereals) was not different between T0 and T3. The distribution of the reasons for refusals is detailed in Table 3. The only reason for children with HOS to refuse a food was because they “did not like” it. Overall, more food refusals occurred with children with ECC than with children with HOS. The difference between the groups was significant for CER and CAR at T0 (*p* ≤ 0.05) but not at T3.

#### 3.6.2. Kinematic Parameters

In children with ECC, chewing frequencies significantly increased for the three foods from T2. The chewing time was significantly lower at T3 compared to T0 for the three foods, while the number of cycles was not statistically different (Table 4). For children with HOS, only the kinematic parameters from the mastication of carrot were significantly different between assessment times. The CAR Frequency significantly increased between T0 and T3, while the CAR chewing time and the CAR number of cycles decreased.

Children with ECC tended to exhibit significantly lower chewing frequencies for the three foods than children with HOS, at both T0 and T3 (except for CHS at T3).

#### 3.6.3. Impairment of Muscular Function

At T0, out of the 16 children with ECC, 12 scored one or more “impairment items” in the clinical assessment of muscular function. The impairment frequency seemed to decrease during the study (7/16 had at least one impairment at T3), but the difference was not significant for any item. No impairment of muscular function was detected in children with HOS at T0. At T3, one child showed impairment of incision and another child had lip incompetence.

The frequencies of “impaired incision”, “lip incompetence”, and “unilateral masticatory pattern” were higher in children with ECC at T0. At T3, there was no statistical difference between either group for any item (Fisher’s exact test, NS).

### 3.7. Masticatory Capability

For the Group with ECC, the D50 values tended to decrease for all foods during the study (Figure 3). For CAR, the D50 values decreased significantly from 4.44 ± 1.05 (T0) to 3.41 ± 0.79 at T2 and 3.48 ± 0.96 at T3. For CHS, the D50 value at T2 (3.88 ± 1.38) was significantly lower than the D50 value at T0 (4.87 ± 1.47). There were no statistically significant differences between timepoints for D50 values of CER. For the Group with HOS, the D50 values seemed to increase between both evaluation times (T0 vs. T3) for all foods, but this difference was significant only for CHS.

The Group with HOS had lower D50 values than the Group with ECC regardless of the type of food. This difference was significant for all foods at T0 but only for CHS at T3 (Figure 3).

### 3.8. Impact of Posterior Teeth Extractions on Chewing Frequency and D50 Values

In the Group with ECC, posterior teeth extractions had no significant impact on the chewing frequencies for the three tested foods (Table 5). All frequencies tended to increase between T0 and T3, with or without extractions.

The D50 values were significantly higher at T0 for CER in children for whom posterior extractions were planned under GA (Table 5). At T3, the D50 values were significantly higher for CAR and CHS in children who had posterior teeth extractions.

## 4. Discussion

This study is the first to assess the impact of dental rehabilitation on children’s masticatory capability, by analyzing the granulometry of natural food boluses collected at the moment of swallowing. It showed that comprehensive treatment under general anesthesia improved masticatory parameters in children with ECC. Indeed, during the twelve-month follow-up period, their chewing frequencies increased and their masticatory capability improved without reaching the same values as children with HOS. In the meantime, their BMI increased significantly, which could reflect that their nutrient intake improved. This study is also the first to record masticatory capability values in children with an evolving healthy oral state.

The data were compared in children with and without ECC because some changes in masticatory parameters were expected to be due to their normal development (dental development state, muscle strength, bone growth, etc.). There are few available datasets related to the evolution of masticatory parameters during childhood. It was suggested that, in children from 2 to 8 years of age, the masticatory time and number of cycles needed for a given natural food tended to decrease throughout childhood [39,40]. Mastication maturity was considered to be reached when the time necessary to chew a certain type of food remains constant across a given age range [39], but methodological differences make it difficult to determine at exactly what age mastication should be fully mature [39,40,41,42]. One recent study from Almotairy et al. (2021) [43] showed no significant variation between groups of subjects ranging from primary to adult dentition while unilaterally masticating soft or hard viscoelastic food models. However, mastication behavior observed under such conditions may be quite different from that adopted with natural foods. A previous systematic review suggested that transition to the adult type of masticatory behavior occurs around the age of 13–14 and that the main influencing factor was dental eruption [43]. In our study conducted on younger children their evolution from primary dentition to early-mixed dentition, the chewing time and the number of masticatory cycles needed for raw carrots decreased significantly over the study period for children with HOS, while the masticatory frequency increased simultaneously for this food. The improvement in sensorimotor control could explain such progress, as described for other functions [43]. Indeed, the positioning of the first permanent teeth occurring at this development stage may increase the food-related sensory information obtained from teeth. The concomitant maturation of the masseter muscle may also help mastication maturation. Moreover, gradual exposure to increasingly diverse and hard textures during childhood supports the maturation of food oral-processing patterns [44,45], along with promoting the development of the orofacial muscular and bone structures. Such anatomical and behavioral adjustments during children’s development may have consequences on the granulometry of natural food boluses after complete mastication (mastication capability), although these evolutions remain unexplored. During the course of this study, food bolus particle size at swallowing in children with HOS tended to increase slightly, possibly because of growth-related changes [46]. To date, normal values of food bolus granulometry during childhood have not been defined, but the D50 values obtained in this study for raw carrot in children with HOS (2.94 ± 0.62 mm at T0 and 3.18 ± 0.73 mm at T3) were consistent with previous data [16]. This value remained under the adult Masticatory Normative Indicator (MNI) of 4 mm for raw carrot, corresponding to the threshold value above which mastication is considered deficient [47]. More physiological data are required to describe the evolution of masticatory capability from primary to permanent dentition during childhood, according to behavioral evolution.

Previous data have strongly suggested that early oral health alterations may negatively affect the development of mastication [16,17] and potentially alter children’s general development. The present study confirmed the impact of ECC on the quality of life of affected children, on the frequency of orofacial dysfunctions, and also on children’s masticatory behavior and mastication capability [5,16,17,21,48]. For the first time, it was shown that comprehensive dental treatment performed under GA succeeded in improving all these functional parameters. Masticatory frequencies for carrot and cereals improved as soon as 6 months after treatment, without reaching those of children without dental alteration after the one-year follow-up. However, the granulometry of food boluses at swallowing did improve after dental treatment for carrots, reaching levels similar to the mastication capability of children with healthy oral state 12 months after GA. The only residual difference between the groups of children at this time was observed for cheese, which might be distorted in a shape more suitable for swallowing, by intra-oral manipulation (tongue, warmth, etc.). Even after dental treatment, and as described in previous studies, children could indeed continue to resort to various masticatory behavior modifications [16,17]. The positive evolution of mastication observed in this study could be partly attributed to the very conservative therapeutic approach used under GA, which may improve mastication sensorimotor control by restoring dental functional contacts. Indeed, children treated with posterior teeth extractions exhibited lower mastication capability one year after treatment compared to children who only received conservative treatments. The D50 value for carrot stayed higher than the adult 4 mm MNI value for children treated with at least one posterior extraction (4.49 mm before treatment versus 4.21 mm one year after treatment), whereas it dropped far below the adult MNI value in children treated only with conservative treatments (4.39 mm before treatment versus 2.75 mm one year after treatment). For cereals, the impact of dental extractions was not observed, probably because children can use saliva to soften this food instead of grinding [17]. It was also expected that the diet of children would diversify after dental rehabilitation. Indeed, the number of food refusals due to difficulties observed during the mastication tests of this study and the food refusals or texture adaptations reported at home decreased. This evolution towards the consumption of more diverse textures could, in turn, contribute towards improving mastication and promoting healthy orofacial growth [49]. In this study, the number of children using a unilateral masticatory pattern, which could lead to posterior crossbite [50], decreased after comprehensive dental treatment. Another study also showed an increased prevalence of posterior balanced occlusion after the treatment of ECC with stainless-steel crowns [51].

In parallel, the height-and-weight status of children with ECC during the follow-up period was modified, with an increase in BMI in this group; about one third of these children reached the overweight or obesity range 12 months after GA. Several other studies obtained similar results with an increase of BMI after dental treatment [24,25,26]. This may be due to the fact that children improve their mastication without changing the quality of their diet. Indeed, one of the risk factors of ECC is daily, frequent, high-sugar food intake, which is also a significant risk factor for obesity [52,53]. Introducing a change in children’s diet has been shown to be challenging [54] and the persistence of feeding cariogenic habits also explains the considerable prevalence of caries relapse in this population [19,22]. Knowing the long-term risk of obesity starting during childhood and the aggravating risk of dental caries relapse, it is necessary to improve nutritional guidance for children treated for ECC and their families [55].

Some limitations to this study do exist. The sample size was small due to loss to follow-up patients (lack of observance from the patients) and this could induce a bias of attrition or self-selection. The large age range (36 to 71 months) could also impact the results. Moreover, children with healthy oral state were evaluated only twice with uneven follow-up periods in between appointments. This group should be evaluated at the same rate as children with ECC in order to pinpoint the exact moment of potential normalization of mastication parameters.

## 5. Conclusions

During the one-year follow-up period after comprehensive dental treatment under GA in children with ECC, masticatory frequencies and mastication capability increased compared to that of children with unaltered oral states. In particular, children without any posterior teeth extractions exhibited better mastication than children with at least one extraction. In parallel, a significant proportion of children became overweight or obese. The links between mastication and nutrition should be addressed in the future.

## Figures and Tables

**Figure 1 ijerph-20-00677-f001:**
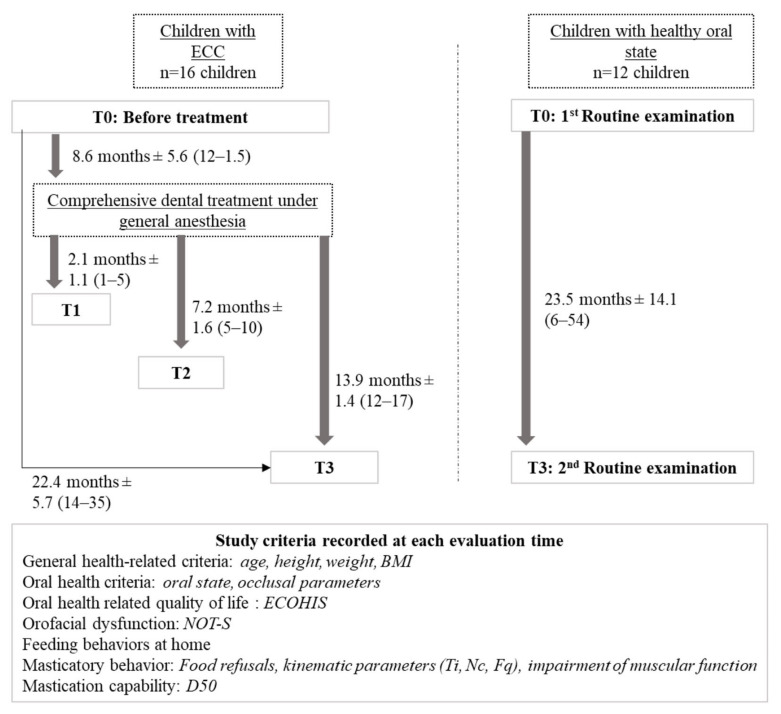
Protocol sequence for both groups of children.

**Figure 2 ijerph-20-00677-f002:**
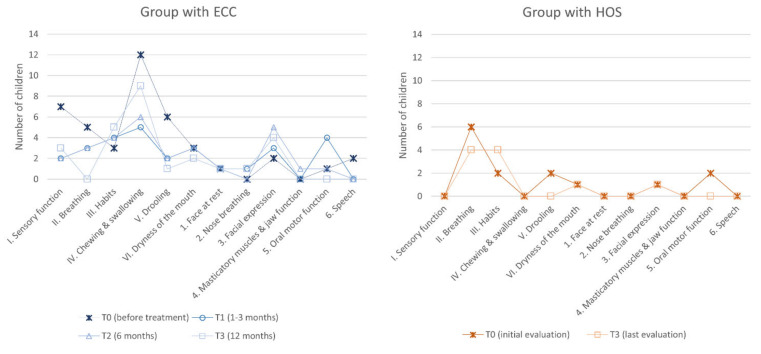
Domain-by-domain orofacial dysfunction frequency throughout the study in both groups of children.

**Figure 3 ijerph-20-00677-f003:**
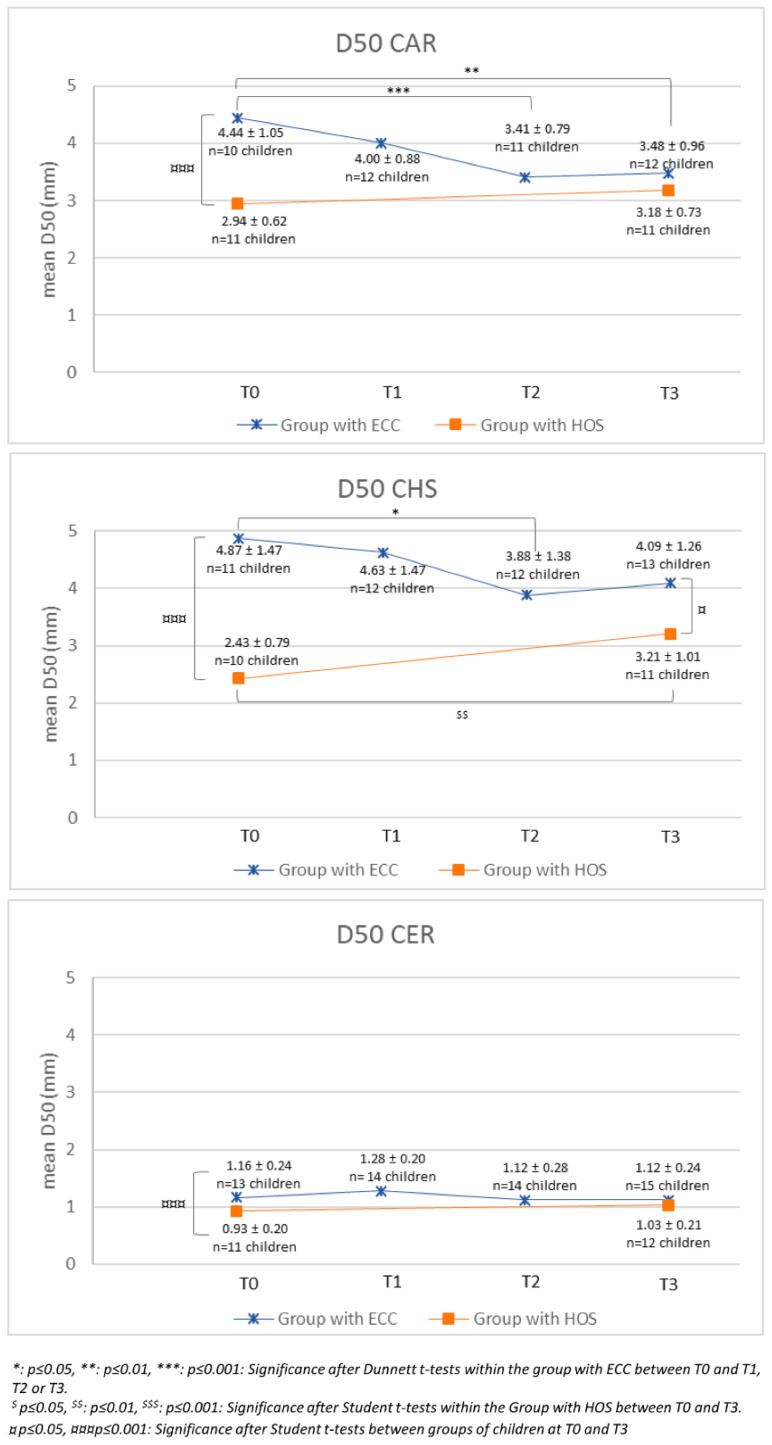
Evolution of mean D50 values at swallowing for the three foods throughout the study period in both groups of children.

**Table 1 ijerph-20-00677-t001:** Comparison of general health data between both groups of children throughout the study.

Criteria	Group with ECC		Group with HOS		Statistical Differences between Groups of Children
	(16 Children)		(12 Children)		
	Evaluation Times	Mean ± SD (Min–Max)	Evaluation Times	Mean ± SD (Min–Max)	Student’s *t*-Test
Age (months)	T0	58.6 ± 8.3 (47–71)	T0	58.4 ± 8.3 (47–71)	NS
	T1	69.3 ± 9.3 (53–82)			
	T2	74.6 ± 9.0 (60–86)			
	T3	80.9 ± 9.5 (66–94)	T3	81.7 ± 16.0 (60–115)	NS
Weight (kg)	T0	17.3 ± 2.2 (13.3–23.3)	T0	19.1 ± 2.4 (15–22.5)	NS
	T1	20.0 ± 3.2 (13.5–25.2)			
	T2	21.7 ± 4.2 (14.5–31.4)			
	T3	23.5 ± 4.9 (15–31)	T3	24.0 ± 6.3 (15–35)	NS
Height (cm)	T0	106.0 ± 5.6 (96.7–116)	T0	110 ± 5.4 (104–120)	
	T1	112.1 ± 6.7 (102–122)			
	T2	115.9 ± 6.0 (104.5–125.5)			
	T3	118.0 ± 6.5 (106–129)	T3	121.6 ± 10.6 (104–138)	NS

NS: no statistical difference (*p* > 0.05).

**Table 2 ijerph-20-00677-t002:** Comparative evolution of oral health criteria in both groups of children throughout the study.

Criteria	Group with ECC		Group with HOS		Differences between Groups
	(16 Children)		(12 Children)		
	Evaluation Time	Mean ± SD	Evaluation Time	Mean ± SD	Student’s *t*-Test
		(Min–Max)		(Min–Max)	
**dmft total score**	**T0**	**12.3 ± 4.6 (3–19)**	**T0**	**0**	***p* ≤ 0.001**
- decayed		11.7 ± 4.4 (3–19)			
- missing		0.2 ± 0.5 (0–2)			
- filled		0.4 ± 1.2 (0–4)			
**PUFA total score**	**T0**	**7.6 ± 4.4 (1–16)**	**T0**	**0**	***p* ≤ 0.001**
- Visible pulp (p)		6.6 ± 4.1 (1–15)			
- Ulceration (u)		**0**			
- Fistula (f)		**0.3 ± 0.4 (0–1)**			
- Abscess (a)		**0**			
**Treatment performed during GA:**					
**Tooth extractions**		**3.5 ± 2.71 (0–8)**		**0**	***p* ≤ 0.001**
- Incisor or canine		2.19 ± 2.10 (0–6)			
- Molar		1.31 ± 1.35 (0–4)			
**Conservative treatment**		**8.19 ± 2.66 (4–13)**		**0**	***p* ≤ 0.001**
- Coronal restauration		2.63 ± 2.0 (0–6)			
- Stainless-steel crown		0.06 ± 0.25 (0–1)			
- Pulpotomy and coronal restauration		0.94 ± 1.48 (0–4)			
- Pulpotomy and stainless-steel crown		4.31 ± 1.82 (2–7)			
- Pulpectomy and coronal restauration		0.13 ± 0.34 (0–1)			
- Pulpectomy and stainless-steel crown		0.13 ± 0.34 (0–1)			
Selective grinding of canine(s)		1 ± 1.79 (0–4)		0	*p* ≤ 0.05
**Number of 1st permanent molar**	T0	0	T0	0	
	T1	1.31 ± 1.62 (0–4)			
	T2	2.06 ± 1.88 (0–4)			
	T3	2.94 ± 1.48 (0–4)	T3	1.83 ± 1.99 (0–4)	NS
**Orofacial dysmorphologies:**		** *n* **		** *n* **	**Fisher’s exact test**
- Dento-maxillary disharmony	**T0**	**0**	**T0**	**0**	**NS**
- Eruption or number of anomalies		**0**		**0**	**NS**
- Sagittal dimension abnormality		**1**		**3**	**NS**
- Transverse dimension abnormality		**3**		**0**	**NS**
- Vertical dimension abnormality		**3**		**5**	**NS**

NS: no statistical difference (*p* > 0.05).

**Table 3 ijerph-20-00677-t003:** Reasons for refusals during the mastication tests throughout the study for both groups of children.

Tested Food	Refusals Reasons	Group with ECC (16)				Group with HOS (12)		Differences between Groups of Children	
		T0	T1	T2	T3	T0	T3	Fisher’s Exact Test	
		*n* Children				*n* Children		T0	T3
CAR	Don’t like	1	1	3	3	1	1		
	Don’t know/Never tried	1	0	1	0	0	0		
	Painful	0	0	0	0	0	0		
	Too difficult to eat	4	2	1	1	0	0		
	Total	6	3	5	4	1	1	*p* ≤ 0.05	NS
CHS	Don’t like	3	2	3	3	2	1		
	Don’t know/Never tried	0	0	1	0	0	0		
	Painful	0	0	0	0	0	0		
	Too difficult to eat	2	1	0	0	0	0		
	Total	5	3	4	3	2	1	NS	NS
CER	Don’t like	1	1	1	1	0	0		
	Don’t know/Never tried	0	0	1	0	0	0		
	Painful	0	0	0	0	0	0		
	Too difficult to eat	2	0	0	0	0	0		
	Total	3	1	2	1	0	0	*p* ≤ 0.05 NS	

NS: no statistical difference (*p* > 0.05).

**Table 4 ijerph-20-00677-t004:** Kinematic parameters according to evaluation times for both groups of children.

Tested Food	Kinematic Parameters	Group with ECC			Group with HOS			Differences between Groups
		Study Time	Number of Children	Mean ± SD	Study Time	Number of Children	Mean ± SD	Student’s *t*-Test
CAR	Ti (s)	T0	10	49.05 ± 35.68	T0	11	33.0 ± 12.95	NS
		T1	13	46.58 ± 31.12				
		T2	11	31.06 ± 12.93				
		T3	12	28.08 ± 11.12 *	T3	11	21.32 ± 5.04 ^$$$^	*p* ≤ 0.05
	Nc (n)	T0	10	44.2 ± 20.17	T0	11	44.5 ± 13.31	NS
		T1	13	53.77 ± 26.50				
		T2	11	43.55 ± 18.20				
		T3	12	38.38 ± 15.14	T3	11	34.18 ± 7.30 ^$$^	NS
	Fq (n/s)	T0	10	1.05 ± 0.31	T0	11	1.43 ± 0.30	*p* ≤ 0.001
		T1	13	1.25 ± 0.28				
		T2	11	1.43 ± 0.33 ***				
		T3	12	1.39 ± 0.22 ***	T3	11	1.64 ± 0.28 ^$^	*p* ≤ 0.01
CHS	Ti (s)	T0	11	21.36 ± 8.10	T0	10	19.65 ± 6.48	NS
		T1	13	22.27 ± 16.66				
		T2	12	15.36 ± 5.94				
		T3	13	14.08 ± 5.63 *	T3	11	16.41 ± 5.24	NS
	Nc (n)	T0	11	23.09 ± 7.49	T0	10	24.1 ± 6.21	NS
		T1	13	21.73 ± 9.91				
		T2	12	20.56 ± 8.55				
		T3	13	19.12 ± 6.94	T3	11	23.5 ± 6.15	*p* ≤ 0.05
	Fq (n/s)	T0	11	1.13 ± 0.22	T0	10	1.36 ± 0.26	*p* ≤ 0.01
		T1	13	1.15 ± 0.39				
		T2	12	1.35 ± 0.26 *				
		T3	13	1.39 ± 0.21 **	T3	11	1.48 ± 0.30	NS
CER	Ti (s)	T0	13	28.04 ± 9.85	T0	12	20.38 ± 5.56	*p* ≤ 0.01
		T1	15	29.9 ± 9.65				
		T2	14	23.57 ± 6.22				
		T3	15	20.43 ± 7.38 **	T3	12	19.29 ± 6.80	NS
	Nc (n)	T0	13	25.04 ± 6.27	T0	12	26.25 ± 6.84	NS
		T1	15	28.33 ± 9.73				
		T2	14	28.36 ± 9.60				
		T3	15	25.87 ± 10.69	T3	12	26.58 ± 6.61	NS
	Fq (n/s)	T0	13	0.95 ± 0.25	T0	12	1.30 ± 0.20	*p* ≤ 0.001
		T1	15	0.99 ± 0.28				
		T2	14	1.20 ± 0.23 ***				
		T3	15	1.27 ± 0.21 ***	T3	12	1.45 ± 0.30	*p* ≤ 0.05

*: *p* ≤ 0.05, **: *p* ≤ 0.01, ***: *p* ≤ 0.001: *p*-value calculated using Dunnett *t*-tests within the Group with ECC between T0 and T1, T2, or T3. ^$^: *p* ≤ 0.05, ^$$^: *p* ≤ 0.01, ^$$$^: *p* ≤ 0.001: *p*-value calculated using Student’s *t*-tests within the Group with HOS between T0 and T3. NS: no statistical difference (*p* > 0.05).

**Table 5 ijerph-20-00677-t005:** Masticatory frequency and D50 values according to posterior teeth extractions in the Group with ECC.

Parameter	Food	Evaluation Time	No Posterior Extractions		≥1 Posterior Extraction	
			Number of Children	Mean ± SD	Number of Children	Mean ± SD
Chewing frequency (Fq)	CAR	T0	5	1.01 ± 0.34	5	1.10 ± 0.29
		T3	6	1.36 ± 0.25	6	1.41 ± 0.20
	CHS	T0	4	1.12 ± 0.15	7	1.13 ± 0.26
		T3	4	1.42 ± 0.32	9	1.38 ± 0.14
	CER	T0	5	0.99 ± 0.19	8	0.93 ± 0.28
		T3	5	1.18 ± 0.18	10	1.32 ± 0.22
D50 (mm)	CAR	T0	5	4.39 ± 0.85	5	4.49 ± 1.26
		T3	6	2.75 ± 0.44	6	4.21 ± 0.77 ***
	CHS	T0	4	4.68 ± 1.33	7	4.98 ± 1.58
		T3	4	3.22 ± 0.56	9	4.48 ± 1.30 **
	CER	T0	5	1.01 ± 0.18	8	1.25 ± 0.24 *
		T3	5	1.12 ± 0.19	10	1.12 ± 0.26

*: *p* ≤ 0.05, **: *p* ≤ 0.01, ***: *p* ≤ 0.001: *p*-value calculated using Student’s *t*-test between children with and without posterior teeth extractions.

## Data Availability

Not applicable.

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
