# Peer review of "Comprehensive Dental Treatment under General Anesthesia Improves Mastication Capability in Children with Early Childhood Caries—A One-Year Follow-Up Study"

_ijerph, 2022, doi:10.3390/ijerph20010677_

Round 1
Reviewer 1 Report
This manuscript reported on a follow-up study investigating mastication capability of children with early childhood caries after comprehensive dental treatment under general aesthesis.
Points to note are:
1. Abbreviation ECC should be introduced in the first paragraph of the Introduction section (line 35), not line 106 in the Methods section.
2. Abbreviation GA was introduced in the Introduction section (line 54). Therefore, it should not be introduced again in the Methods section (line 107).
3. Discussion section should include limitations of the study, taking into account sources of potential bias or imprecision.
4. Reference 30: The format in citating a book is incorrect. There is no need to introduce the abbreviation WHO.
5. Reference 30: ‘Organisation’ should be ‘Organization’
Author Response
Dear reviewer,
Thank you for your comments. Please find our responses below:
- Abbreviation ECC should be introduced in the first paragraph of the Introduction section (line 35), not line 106 in the Methods section.
The abbreviation was added in the introduction section
- Abbreviation GA was introduced in the Introduction section (line 54). Therefore, it should not be introduced again in the Methods section (line 107).
The abbreviation was removed.
- Discussion section should include limitations of the study, taking into account sources of potential bias or imprecision.
A paragraph on study limitations was added in the discussion section line 542 to 548.
Some limitations to this study do exist. Sample size is small due to loss to follow-up patient (lack of observance from the patients) and this could induce a bias of attrition or self-selection. The large age range (36 to 71 months) could also impact the results. Moreover, children with healthy oral state were evaluated only twice with uneven follow-up periods in between appointments. This group should be evaluated at the same rate than children with ECC in order to pinpoint the exact moment of potential normalization of mastication parameters.
- Reference 30: The format in citating a book is incorrect. There is no need to introduce the abbreviation WHO.
The reference was changed accordingly.
- Reference 30: ‘Organisation’ should be ‘Organization’
The reference was modified according to the comment. Thank you.
Reviewer 2 Report
The study is well organized and written
-Abstract: please remove: “for the first time”
-Line 155, “ The tests were stopped at will”- do you mean “as well”
- I believe Socioeconomic status may have effect on your topic, did you reported any change in that “relocation – change Schools – area”
- As you mentioned in the discussion, dental extractions may have higher impact on your results, Did you tried to subgroup analysis based on that?
- Please add reference to sentience line 521-523
- Conclusion, please remove “this is the first study…..” –you already mentioned that in the discussion
-Please make the conclusion more clear and make the conclusion consistent between the abstract and manuscript .
Author Response
Dear reviewer,
Thank you for your comments. Please find below our responses:
The study is well organized and written
Thank you!
-Abstract: please remove: “for the first time”
It was removed.
- Line 155, “ The tests were stopped at will”- do you mean “as well”
No, we do mean “at will”, meaning that the mastication tests were stopped immediately when children expressed the need to. To clarify, we changed the sentence to: “The children were encouraged to express any difficulty upon refusal or uneasiness with the food samples, and the tests were stopped at any time”.
- I believe Socioeconomic status may have effect on your topic, did you reported any change in that “relocation – change Schools – area”
We agree that socioeconomic status has an impact on caries prevalence. Children with ECC are known to belong mostly in families with low socioeconomic status. This could influence the quality and type of children’s diet and limit the mastication progress, even after treatment. This parameter is inherent to the population characteristics, and we did not ask about socioeconomic status during the course of our study.
Moreover, treatment was not dependent on the socioeconomic status of the patient, as general anesthesia is completely covered by the state health insurance.
- As you mentioned in the discussion, dental extractions may have higher impact on your results, Did you tried to subgroup analysis based on that?
The subgroup analysis is indeed presented in paragraph 3.8.
- Please add reference to sentience line 521-523
The reference [17] was added line 520. The diversification of diet after rehabilitation is our hypothesis, and is not supported by any known reference (to our knowledge).
- Conclusion, please remove “this is the first study…..” –you already mentioned that in the discussion
This has been removed as followed: “This is the first study to assessed masticatory capability….”
- Please make the conclusion more clear and make the conclusion consistent between the abstract and manuscript
We changed the conclusion according to your expectation.
During the one-year follow-up period after comprehensive dental treatment under GA in children with ECC, masticatory frequencies and mastication capability increased compared to that of children with unaltered oral state. In particular, children without any posterior teeth extractions exhibited better mastication than children with at least one extraction. In parallel, a significant proportion of children became overweight or obese. Links between mastication and nutrition should be addressed in the future.